# Dispersion Stability of 14 Manufactured Nanomaterials for Ecotoxicity Tests Using *Raphidocelis subcapitata*

**DOI:** 10.3390/ijerph19127140

**Published:** 2022-06-10

**Authors:** Seung-Hun Lee, Kiyoon Jung, Won Cheol Yoo, Jinwook Chung, Yong-Woo Lee

**Affiliations:** 1Department of Chemical and Molecular Engineering, Hanyang University, 55 Hanyangdaehakro, Sangrok-gu, Ansan 15588, Korea; lck2394sin@hanyang.ac.kr (S.-H.L.); jung940819@naver.com (K.J.); wcyoo@hanyang.ac.kr (W.C.Y.); 2Department of Applied Chemistry, Center for Bionano Intelligence Education and Research, Hanyang University, Ansan 15588, Korea; 3R&D Center, Samsung Engineering Co., Ltd., 41 Maeyoung-ro, 269 Beon-gil, Youngtong-gu, Suwon 16523, Korea

**Keywords:** dispersion, ecotoxicity, manufactured nanomaterial, *Raphidocelis subcapitata*

## Abstract

The development of nanotechnology has increased concerns about the exposure of ecosystems to manufactured nanomaterials, the toxicities of which are now being researched. However, when manufactured nanomaterials are mixed with algae in a culture medium for ecotoxicity tests, the results are vulnerable to distortion by an agglomeration phenomenon. Here, we describe a dispersion method commonly applicable to ecotoxicity tests for the 14 types of manufactured nanomaterials specified by the Organisation of Economic Co-operation and Development’s Sponsorship Programme, namely aluminum oxide (Al_2_O_3_), carbon black, single-walled carbon nanotubes (SWCNTs), multi-walled carbon nanotubes (MWCNTs), cerium oxide (CeO_2_), dendrimers, fullerene, gold (Au), iron (Fe), nanoclays, silver (Ag), silicon dioxide (SiO_2_), titanium dioxide (TiO_2_), and zinc oxide (ZnO). The type of dispersant, sonication time, and stirring speed were carefully considered. Consequently, 1500 mg/L of gum arabic was selected as a dispersant; for sonication time, 1 h was selected for dendrimers, 2 h for SiO_2_, 24 h for SWCNTs and Au, and 4 h for the other nanomaterials. Dispersion stability was achieved for all materials at a stirring speed of 200 rpm. To verify the effect of this dispersion method on ecotoxicity tests, toxicity was measured through cell counts for SWCNTs and TiO_2_ using *Raphidocelis subcapitata*. The half-maximal effective concentrations (EC_50_) were 18.0 ± 4.6 mg/L for SWCNTs and 316.6 ± 64.7 mg/L for TiO_2_.

## 1. Introduction

Nanotechnology, now a core field of advanced science in the 21st century, is being applied in a wide variety of areas and has contributed to the acceleration of industrial development. Nanotechnology has helped improve quality of life in many respects; however, the increasing use of manufactured nanomaterials has led to concerns about the risks they pose to human health and the environment [1]. Research programs are now underway in multiple countries to evaluate the potential risks and compare the toxicity of various nanomaterials used in major applications [2,3]. As part of this effort, the Organisation for Economic Co-operation and Development (OECD) selected 14 types of manufactured nanomaterials—Al_2_O_3_, carbon black, SWCNTs, MWCNTs, CeO_2_, dendrimers, fullerene, Au, Fe, nanoclays, Ag, SiO_2_, TiO_2_, ZnO—and is working with several countries to test the toxicity of these materials [2].

Toxicity tests can involve technical problems such as agglomeration, and it is often necessary to establish dispersion stability for useful test results. Keller et al. [4] described the agglomeration of TiO_2_ in water, and Sillanpää et al. [5] found, during an experiment using TiO_2_ particles originally measuring 20 nm, that particle size increased to 1000 nm only 60 min after stirring was initiated. This is likely due to the zeta potential on the surface of the nanomaterial and the electrostatic attraction between ions in water [6]. Such changes in the size of manufactured nanomaterials are inevitable in a culture medium used for ecotoxicity tests involving aquatic life. Changes in the size of manufactured nanomaterials may also affect the toxicity mechanism itself. Pan et al. [7] reported that the toxicity trend of Au appeared to vary depending on particle size. When Au particles 5, 15, and 40 nm in size were exposed to *Scrobicularia plana*, the toxicity mechanism differed according to the size of the material. Similarly, the amount of Au that accumulated in *S. plana* increased depending on the size of the Au particles, and comparable trends have been reported for other manufactured nanomaterials [8,9]. In ecotoxicity studies of manufactured nanomaterials, it is therefore crucial to disperse nanoparticles at constant sizes. Previous studies on the toxicity of manufactured nanomaterials used various methods to stably disperse their particles. For example, Hartmann et al. [10] conducted sonication for 10 min immediately before dispersing a 250 ppm TiO_2_ suspension, kept it in a dark place at 5 °C until the test was finished, and then conducted sonication again for 10 min. In a later study, Hartmann et al. [11] conducted sonication in a water bath for 10 min to create a TiO_2_ suspension of 1000 ppm. Although dispersion could be maintained at the early stage of the test, it was not clear whether manufactured nanomaterials remained dispersed until the end of testing. Ecotoxicity tests for pure gold are rare, and there is a need to improve dispersion performance by modifying the surface with other functional groups, such as citrate, usually on the outside [12]. However, because such tests examine the toxic effects of the functional group or coated materials and not those of pure gold, it is unclear whether the toxicity of the corresponding manufactured nanomaterial itself is being measured.

If the dispersion method differs for every material, comparing the toxicity of nanomaterials becomes difficult. This problem also applies to factors other than the dispersion method. It is difficult to compare toxicity between materials because each study tests different materials and uses different culture medium compositions, different particle sizes, and different analytical methods. In previous tests of carbon black, Knauer et al. [13] conducted ecotoxicity studies on *Raphidocelis subcapitata* that, unlike other studies on carbon black, neither considered the carbon black particle size nor examined its toxicity. Canesi et al. [14] did consider the particle size of carbon black but used *Mytilus galloprovincialis* rather than algae as a test model. It is not easy to compare these results with those of conventional tests that use algae or water fleas. A dispersion method applicable across different material types is required to compare the toxicities of various types of manufactured nanomaterials. Here, we describe a dispersion method applicable to ecotoxicity tests using *R. subcapitata*, the official test species specified by the OECD for tests of the 14 manufactured nanomaterials included in its Sponsorship Programme for the Testing of Manufactured Nanomaterials. These results may be helpful in establishing dispersion conditions for the toxicity testing of various manufactured nanomaterials.

## 2. Materials and Methods

### 2.1. Selection of Manufactured Nanomaterials

The Sponsorship Programme selected 14 species of manufactured nanomaterials, defining the material names but not the particle sizes of the materials. We therefore based the selection of the manufactured nanomaterials for our studies on their frequency of use in industrial and research applications. The manufactured nanomaterials to be dealt with in this study were selected by investigating the size and type of manufactured nanomaterials commonly used in industry and research. From KoreaNano (Gwangmyeong, Korea), we purchased the following manufactured nanomaterials: Al_2_O_3_ (20 nm, gamma, 99%), CeO_2_ (30 nm, >99.9%), Au (15 nm, >99.9%), Fe (25 nm, >99%), Ag (20 nm, >99.9%), SiO_2_ (15–20 nm, >99.5%), and ZnO (35–45 nm, 99%). From Sigma-Aldrich (St. Louis, MO, USA) we purchased MWCNTs (D: 5–50 nm, L: 5–15 μm, >98% carbon basis), dendrimers (10 nm, G4-PAMAM dendrimer, 10 wt% in methanol), nanoclays (100 nm, bentonite), and TiO_2_ (21 nm, P25, >99.5%). Carbon Nanotech (Pohang, Korea) supplied carbon black (30 nm, >99%) and SWCNTs (D: 1–2 nm, L: −10 μm, >95%), and Alfa Aesar (Haverhill, MA, USA) supplied fullerene (30 nm, C_60_, 99%). The uses and properties of nanomaterials are added to Table 1 [8,15,16,17,18,19,20,21,22,23].

### 2.2. Selection of Dispersion

We conducted an experiment to select a dispersant and its optimal concentrations for alleviating the dispersion problem. The three reagents that we used as dispersants, namely spray-dried gum arabic (Sigma-Aldrich, St. Louis, MO, USA), polyoxyethylene castor oil (HCO-40, KAO, Tokyo, Japan), and polyvinylpyrolidone (quality level 200, Sigma-Aldrich, St. Louis, MO, USA), were known to have the lowest toxicity levels of the available options [24,25,26,27]. Based on previous findings regarding the dispersion of manufactured nanomaterials, we selected the following concentrations: 1000–10,000 mg/L for gum arabic, 1000–10,000 mg/L for polyoxyethylene castor oil, and 50–400 mg/L for polyvinylpyrrolidone [24,25,26,27]. Through preliminary testing, we selected six manufactured nanomaterials (Al_2_O_3_, carbon black, SiO_2_, nanoclays, TiO_2_, and ZnO) that clearly aggregated and precipitated when dispersants were not used. We selected a concentration of manufactured nanomaterials that could reach EC_100_ according to previous studies [15,16,17,18,19,20,21,22]. The selected concentrations of the manufactured nanomaterials were 500 mg/L of Al_2_O_3_, 1000 mg/L of carbon black, 500 mg/L of SiO_2_, 1000 mg/L of nanoclays, 2000 mg/L of TiO_2_, and 100 mg/L of ZnO. The manufactured nanomaterials and dispersants were prepared in a 100 mL culture medium prepared according to OECD TG 201 and vigorously shaken for 5 min in a graduated cylinder [28]. The shaking was stopped for 40 s before checking the culture medium’s dispersion stability. The dispersion stability was determined as the average value of three repeated measurements of the difference in turbidity between the upper and lower layers (1/3 and 2/3) of the graduated cylinder using a turbidimeter (2100Q, HACH Co., Loveland, CO, USA). In addition, we examined the effect of the dispersant on algae by conducting an ecotoxicity test for *R. subcapitata*. The conditions of all experiments were in accordance with OECD TG 201 (see Table 2) [28]. Also, the concentration of gum arabic was 750–12,000 mg/L, and the common ratio of the gum arabic concentration was 2.

### 2.3. Selection of Sonication Time

The experiment described in the preceding section allowed us to select a dispersant that worked effectively with the manufactured nanomaterials used in this study. As it was difficult to completely disperse the nanomaterials by adding dispersants, we used an ultrasonic crusher to improve dispersion performance. However, extended sonication introduces the possibility of crushing the manufactured nanomaterials [29]. We therefore made several attempts to determine a minimum sonication time that could achieve optimal dispersion without breaking the nanomaterials under the selected dispersant conditions. We added 100 mL of the OECD medium in which 1500 mg/L of gum arabic was dissolved to each of the following quantities of the manufactured nanomaterials that noticeably aggregated and precipitated without a dispersant: 500 mg/L of Al_2_O_3_, 1000 mg/L of carbon black, 500 mg/L of SiO_2_, 1000 mg/L of nanoclays, 2000 mg/L of TiO_2_, and 100 mg/L of ZnO. Each culture medium was then exposed to ultrasonic frequencies for 10, 30, 60, 120, 240, and 480 min. The temperature of the culture medium was maintained at 20–25 °C using an ice bath, and changes in temperature were monitored throughout the whole experiment period. 10 mL was collected from the upper and lower layers for each sonication time, and the difference in turbidity was measured three times.

### 2.4. Selection of Stirring Speed

To maintain a specific degree of dispersion throughout the test, we investigated an optimal stirring speed, as specified in OECD Test No. 201 for the whole 72 h test period. To determine whether the optimal dispersion conditions could be applied to other manufactured nanomaterials, we performed this test for 13 of the 14 manufactured nanomaterials listed in OECD’s Sponsorship Programme for the Testing of Manufactured Nanomaterials; dendrimers, which easily disperse in water because they belong to the highly water-soluble NH_4_^+^ functional group, were excluded. We selected the following concentrations of the manufactured nanomaterials at which we expected their respective culture media to exhibit a EC_100_: 500 mg/L for Al_2_O_3_, 1000 mg/L for carbon black, 100 mg/L for SWCNTs, 500 mg/L for MWCNTs, 500 mg/L for CeO_2_, 100 mg/L for fullerene, 500 mg/L for Au, 500 mg/L for Fe, 1000 mg/L for nanoclays, 20 mg/L for Ag, 500 mg/L for SiO_2_, 2000 mg/L for TiO_2_, and 100 mg/L for ZnO. Next, we transferred each of the culture media to a 250 mL Erlenmeyer flask, which was shaken for 72 h at 100–200 rpm in a shaking incubator (NB-205VQ, N-Biotek, Bucheon, Korea). Because collecting samples from a 250 mL Erlenmeyer flask is difficult due to the fact that the level of the solution is low (leaving a small distance between the top and bottom layers of the solution), it is crucial to verify dispersion stability by measuring turbidity three times hourly at a half point of the solution and then comparing it with the initial turbidity. If dispersion stability is not maintained, the stirring speed should be increased and the dispersion stability re-evaluated.

### 2.5. Measurement of Shape and Size Distribution

Effectively determining the need for pretreatment to improve dispersion requires observing the dispersed shapes of the manufactured nanomaterials in the ecotoxicity test solution and comparing them with their original shapes. For this purpose, we used a scanning electron microscope (SEM; Hitachi S-4800, Hitachi, Tokyo, Japan) and a transmission electron microscope (TEM; JEM-2100F, JEOL, Tokyo, Japan). In addition, we used a zeta potential analyzer (ZetaPALS, Brookhaven Instruments, New York, NY, USA) to examine the cause of the dispersion trends of the manufactured nanomaterials and to measure the zeta potential of the manufactured nanomaterials in the algal culture medium with and without the gum arabic. Concentrations at which the toxicity of the manufactured nanomaterial was expected to represent EC100 were prepared, as follows: 500 mg/L for Al_2_O_3_, 1000 mg/L for carbon black, 100 mg/L for SWCNTs, 500 mg/L for MWCNTs, 500 mg/L for CeO_2_, 100 mg/L for fullerene, 500 mg/L for Au, 500 mg/L for Fe, 1000 mg/L for nanoclays, 20 mg/L for Ag, 500 mg/L for SiO_2_, 2000 mg/L for TiO_2_, and 100 mg/L for ZnO. The pH of the culture medium was 6.4, and it was confirmed that it was maintained between 6.2 and 7.1 even after the dispersion of the manufactured nanomaterial.

### 2.6. Evaluation of Dispersion Stability

To verify the possibility of maintaining dispersion stability, we conducted an ecotoxicity test for SWCNTs and TiO_2_ under previously established experimental conditions. The test for *R. subcapitata* was performed by referring to OECD Test No. 201: Freshwater Algae and Cyanobacteria, Growth Inhibition Test [28]. All glasses and distilled water used in this test were sterilized at 120 °C for 15 min to minimize the risk of contamination. After inoculating *R. subcapitata* to achieve an initial inoculation concentration of 1 × 10^4^ cells/mL approximately three days before starting the test, the algae were pre-cultured and used in the main test at the exponential growth stage. For the five treatment groups, excluding the control group, a common ratio below 3.2 at each nanomaterial concentration was maintained according to the OECD test method. The algae exposed to the test materials were put in a shaking incubator, and the temperature and luminous intensity were maintained in accordance with the culture conditions during the experiment. Measurements were carried out by the cell counting method using an optical microscope (NBS-80T, Samwon, Yeongcheon, Korea), and the results were tabulated by classifying the cell concentrations of the treatment and control groups by the measurement time and the concentration of the test material. The growth inhibition rate (percent inhibition of average specific growth rate) was determined by the ratio of the cell count of the control group to that of the treatment group. The half-maximal effective concentration (EC_50_) with a 95% confidence interval was calculated using the log-probit function in MedCalc (a toxicity-calculation software package) based on the growth inhibition rate, which was obtained through our own experimental efforts. This EC_50_ measurement was compared with the EC_50_ measurements reported in previous studies to verify the possibility of using the dispersion stability obtained in this study in actual ecotoxicity tests.

## 3. Results and Discussion

### 3.1. Selection of Dispersant

To obtain the highest concentration at the point where the toxicity of the target manufactured nanomaterials becomes evident, we selected three types of dispersants based on previous studies of this subject. We then mixed the OECD medium and verified the culture medium’s dispersion stability by comparing it with the turbidity values for the top and bottom layers of the culture medium for Al_2_O_3_, carbon black, nanoclays, SiO_2_, TiO_2_, and ZnO (Figure 1). As a result of comparing the three types of dispersants, we selected gum arabic at a concentration of 1500–3000 mg/L to secure dispersion stability. We found that gum arabic at that concentration could more effectively disperse manufactured nanomaterials compared with other dispersants, increasing dispersion by up to 84%.

The results of the ecotoxicity test conducted using gum arabic as described above are shown in Figure 2. A negative toxicity result was obtained at concentrations between 1500 and 6000 mg/L, and growth inhibition was discovered at higher concentrations. This indicates that when gum arabic is used below a certain concentration, it contributes to its own growth in the culture medium; when the concentration is too high, the gum arabic interferes with the growth of algae. Gum arabic therefore stabilizes the dispersion of manufactured nanomaterials, but the amount of gum arabic should be minimized to limit its effect on the algae. We chose 1500 mg/L as the optimal amount of dispersant to minimize its adverse effects on algae at concentrations between 1500 and 3000 mg/L, in which optimal dispersion performance can be maintained.

### 3.2. Selection of Optimal Sonication Time

After achieving optimal dispersion efficiency for all manufactured nanomaterials when 1500 mg/L of gum arabic was added to the culture medium, we made an attempt to maximize dispersion efficiency by setting an optimal sonication time, which is the point at which the difference in turbidity between the top and bottom layers becomes minimal. The experiment we conducted showed that the optimal sonication time varied by material (Figure 3). The turbidity difference was at its minimum at 2 h for SiO_2_ and at 4 h for Al_2_O_3_, carbon black, nanoclays, TiO_2_, and ZnO. This result suggests that, for the dispersion of manufactured nanomaterials, a different sonication time must be specified for the 14 types of manufactured nanomaterials. The turbidity difference between the top and bottom layers of the culture medium was minimized at 1 h for dendrimers, 2 h for SiO_2_, 24 h for SWCNTs and Au, and 4 h for other materials. These times were selected as the optimal sonication times.

### 3.3. Selection of Optimal Stirring Speed

Regarding the optimal dispersion conditions for manufactured nanomaterials, 72 h dispersion stability at 100 rpm was observed by applying the optimal sonication time for 13 of the 14 manufactured nanomaterials included in the OECD’s Sponsorship Programme for the Testing of Manufactured Nanomaterials (dendrimers were excluded because they belong to the water-soluble NH_4_^+^ functional group, which easily disperses in water). A turbidity reduction of up to 10% for 72 h was considered a reliable indicator of dispersion stability, and we concluded that dispersion stability was achieved and maintained in our tests for carbon black, MWCNTs, CeO_2_, nanoclays, Ag, SiO_2_, TiO_2_, and ZnO. However, for Al_2_O_3_, SWCNTs, fullerene, Au, and Fe, a reduction in turbidity of more than 10% was evident after 72 h. We therefore increased the stirring speed to 200 rpm, after which we were able to confirm that turbidity decreased to less than 10% for 72 h in every test material except Au. We concluded that dispersion stability was obtained within the test period (Table 3). In our dispersion stability tests, Au showed a turbidity reduction rate of 10.4% at 200 rpm, exceeding our own criterion of a 10% reduction. As the difference of 0.4% was relatively small, we considered Au to have achieved dispersion stability within the error range. The improvements in dispersion stability brought about by changing the rpm or stirring speed appear to be related to the density of the material and the rotational kinetic energy applied by the shaking incubator. Materials of high density settle due to their inherent weight, even when dispersed at 100 rpm, but at faster speeds, it becomes easier for the particles in the solution to float due to increased rotational kinetic energy. This phenomenon is consistent with the fact that Au, with the highest density (19.3 g/m^3^) of the tested nanomaterials, precipitated severely, resulting in a reduction in turbidity reduction to 65.2% at 100 rpm. However, the turbidity reduction rate improved to 10.4% when the stirring speed was increased to 200 rpm. Finally, we performed an ecotoxicity test to verify whether the selected stirring speed of 200 rpm affected the algae. The result showed that, at such a speed, algal growth after 72 h was 58 times the initial algae inoculation concentration. This was higher than the 16 times growth after the initial algae inoculation, which is the criterion of the OECD test method. We therefore concluded that a stirring speed of 200 rpm was the optimal shaking condition for the ecotoxicity test.

### 3.4. Change in Shape and Size Distribution

We compared the shape of particles after dispersing manufactured nanomaterials in ethanol during sonication (before treatment) with the shape of particles in a pretreatment condition during the ecotoxicity test (after treatment). Most of the manufactured nanomaterials were observed with an SEM, but we observed dendrimers with a TEM because they could not be observed with an SEM. Platinum coating for SEM measurement was performed for only 1 min to minimize its effect at a measurable level, and the same magnification was used for measurements before and after treatment. The shapes of each material before and after treatment are summarized in Figure 4. The sizes of the manufactured nanomaterial particles increased by an average factor of 21 compared with their sizes before treatment. The reason for this agglomeration phenomenon can be explained by the Derjaguin, Landau, Verwey, and Overbeek (DLVO) theory, which describes the colloid phenomenon in a liquid [30]. According to this theory, the dispersion stability of manufactured nanomaterials in a liquid is determined by electrical repulsion between the manufactured nanomaterials and whether the Van Der Waals force is stable. Electrical repulsion between manufactured nanomaterials is caused by their respective surface charges, which vary depending on the unique properties of the manufactured nanomaterial and the nature of the surrounding medium. Bae et al. [31] reported that unique properties appeared through the electric double layer of manufactured nanomaterials. If the electric double layer is thick, the dispersion of manufactured nanomaterials in a solution is stable; if it is thin, agglomeration of nanomaterials can occur. The thickness of the electric double layer can be inferred by measuring the zeta potential. The closer the magnitude of the zeta potential is to zero, the thinner the electric double layer. Here, the magnitude of the zeta potential is affected by the composition of the solution, the pH, the ionic strength, and the surface charge of the material. In general, it is known that aggregation occurs within ±5 mV of the zeta potential, dispersion is unstable within ±10–±30 mV, and dispersion is stably maintained when it is over ±30 mV [31]. To verify the dispersion stability of manufactured nanomaterials, we measured the zeta potential of each material used in the culture medium (pH 6.4). The results showed that the surface zeta potential of every material except ZnO was negative, and before adding gum arabic, all materials except Fe, nanoclays, and Ag showed a zeta potential below ±30 mV, maintaining an unstable or non-dispersion state. These zeta potential values are estimated to be an effect of the anions in water for ZnO and metal cations in water for other materials, which made it impossible to maintain the dispersion stability of the manufactured nanomaterials without dispersants. However, after adding gum arabic, the zeta potential increased. In particular, the zeta potential of Al_2_O_3_, which had a dispersion problem, was −21 mV, with Au exhibiting a zeta potential of −13 mV and all the other materials showing zeta potentials higher than ±30 mV, achieving dispersion stability (Table 4). This phenomenon can be attributed to the fact that gum arabic wraps around the manufactured nanomaterials when it is added for dispersion stability. This increases the respective zeta potential, and the dispersion stability appears to be improved by the electrostatic attraction between them. In fact, the absolute value of the zeta potential increased by 118.1%, on average, compared with zeta potentials before the addition of gum arabic.

### 3.5. Evaluation of Dispersion Stability

We verified the applicability of the dispersion conditions established to this point to the actual ecotoxicity tests we conducted for manufactured nanomaterials. We are therefore confident in the validity of the ecotoxicity tests performed for SWCNTs and TiO_2_, for which ecotoxicity studies using *R. subcapitata* were previously conducted, and we found that our EC_50_ measurements were comparable with those reported in the existing literature. In particular, after we exposed our algae test species to the dispersed solutions of SWCNTs and TiO_2_, we accurately determined the growth inhibition rate using the cell counting method for both the control and treatment groups at 0, 24, 48, and 72 h (Figure 5). Our calculated results for EC_50_ were 18.0 ± 4.6 mg/L for SWCNTs and 316.6 ± 64.7 mg/L for TiO_2_. This toxicity level was approximately 40% higher than that of the EC_50_ level of 30.0 mg/L for SWCNTs reported in previous studies. In addition, TiO_2_ showed an approximately 24% greater toxicity than the EC_50_ value of 415 mg/L reported previously [32,33]. These differences may have been caused by differences in the dispersion method of the nanomaterials. When Sohn and Hund-Rinke used bovine serum albumin (BSA) as a dispersant, they considered the possibility that the observed toxicity levels fell because the manufactured nanomaterials formed BSA-nanoparticles by adsorbing to BSA [34]. This property of BSA has been reported in the past, and the phenomenon of lowering the toxicity level of chemicals, including that of manufactured nanomaterials, has been reported [35,36,37]. Authors reported EC50 values measured by the same dispersion method [38]: Al_2_O_3_: 53.9 ± 10.2 mg/L, carbon black: 4.8 ± 2.2 mg/L, MWCNTs: 9.0 ± 1.5 mg/L, CeO_2_: 26.2 ± 4.6 mg/L, dendrimers: 24.6 ± 3.9 mg/L, fullerene: 395.3 ± 51.6 mg/L, Au: 352.6 ± 137.3 mg/L, Fe: 64.3 ± 10.6 mg/L, nanoclays: 11,245.6 ± 2342.3 mg/L, Ag: 0.3 ± 0.04 mg/L, SiO_2_: 211.3 ± 22.7 mg/L, and ZnO: 0.06 ± 0.04 mg/L. These ecotoxicity test results suggest that applying the dispersion method for nanomaterials proposed in this study to the 14 manufactured nanomaterials included in the OECD’s Sponsorship Programme for the Testing of Manufactured Nanomaterials poses no problems. This method could also be applied to ecotoxicity tests of manufactured nanomaterials other than the 14 types targeted in this study.

## 4. Conclusions

This study aimed to investigate the dispersion stability of manufactured nanomaterials in a culture medium for the toxicity test method using *R. subcapitata* as the OECD official test species. However, for toxicity tests using *R. subcapitata* as specified in OECD Test No. 201, a uniform dispersion method for manufactured nanomaterials is required because agglomeration and precipitation of the manufactured nanomaterials can occur due to the effect of cations present in the culture medium. To this end, the optimal dispersant and its injection concentration, the optimal sonication time, and the optimal stirring speed must be carefully chosen and calibrated. First, regarding the optimal dispersant and its injection concentration, 1500 mg/L of gum arabic was selected in consideration of the dispersing performance and the effect of the dispersant on algae. For the optimal sonication time, our test results indicated that a different sonication time should be used for each material. We found that an optimal dispersion efficiency was obtained at 1 h for dendrimers, 2 h for SiO_2_, 24 h for SWCNT and Au, and 4 h for the rest of the OECD-listed manufactured nanomaterials, which suggests that these values are widely applicable. Regarding the optimal stirring speed, we found that dispersion stability could be maintained for 72 h when the material was stirred at 200 rpm. We also verified that dispersion stability could be achieved by changing the zeta potential of the manufactured nanomaterials. The zeta potential increased by 118.1% on average compared to before dispersion treatment, confirming the stability of the dispersion. The optimal dispersion conditions for ecotoxicity tests of manufactured nanomaterials and the proposed dispersion method should be applicable to ecotoxicity tests for manufactured nanomaterials other than the 14 types listed in the OECD’s Sponsorship Programme for the Testing of Manufactured Nanomaterials.

## Figures and Tables

**Figure 1 ijerph-19-07140-f001:**
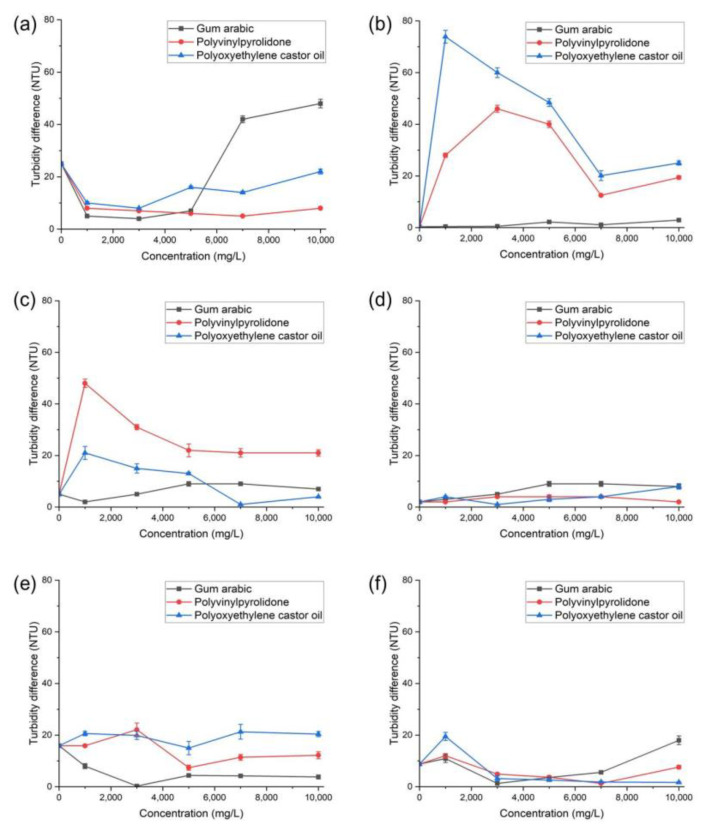
Turbidity difference values of manufactured nanomaterials by dispersant: (**a**) Al_2_O_3_, (**b**) carbon black, (**c**) nanoclays, (**d**): SiO_2_, (**e**) TiO_2_, and (**f**) ZnO.

**Figure 2 ijerph-19-07140-f002:**
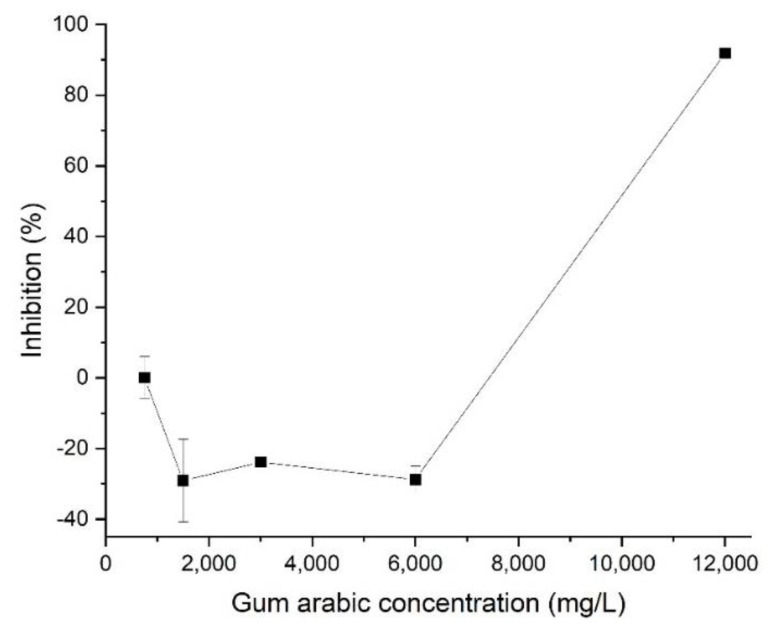
Inhibition of gum arabic in a 72 h exposure test.

**Figure 3 ijerph-19-07140-f003:**
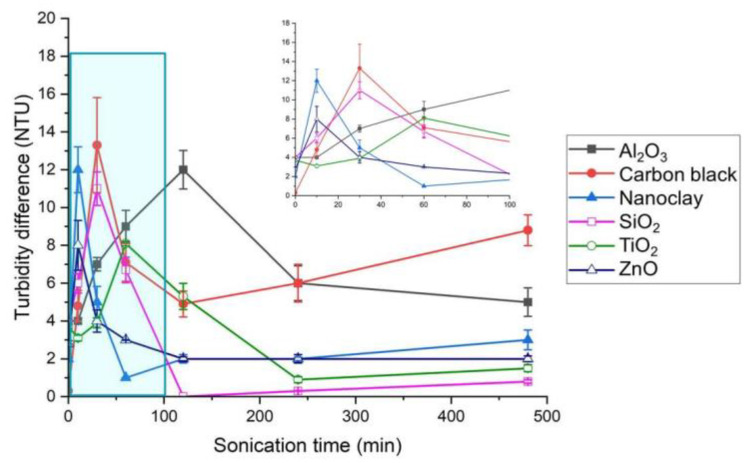
Turbidity difference by sonication time.

**Figure 4 ijerph-19-07140-f004:**
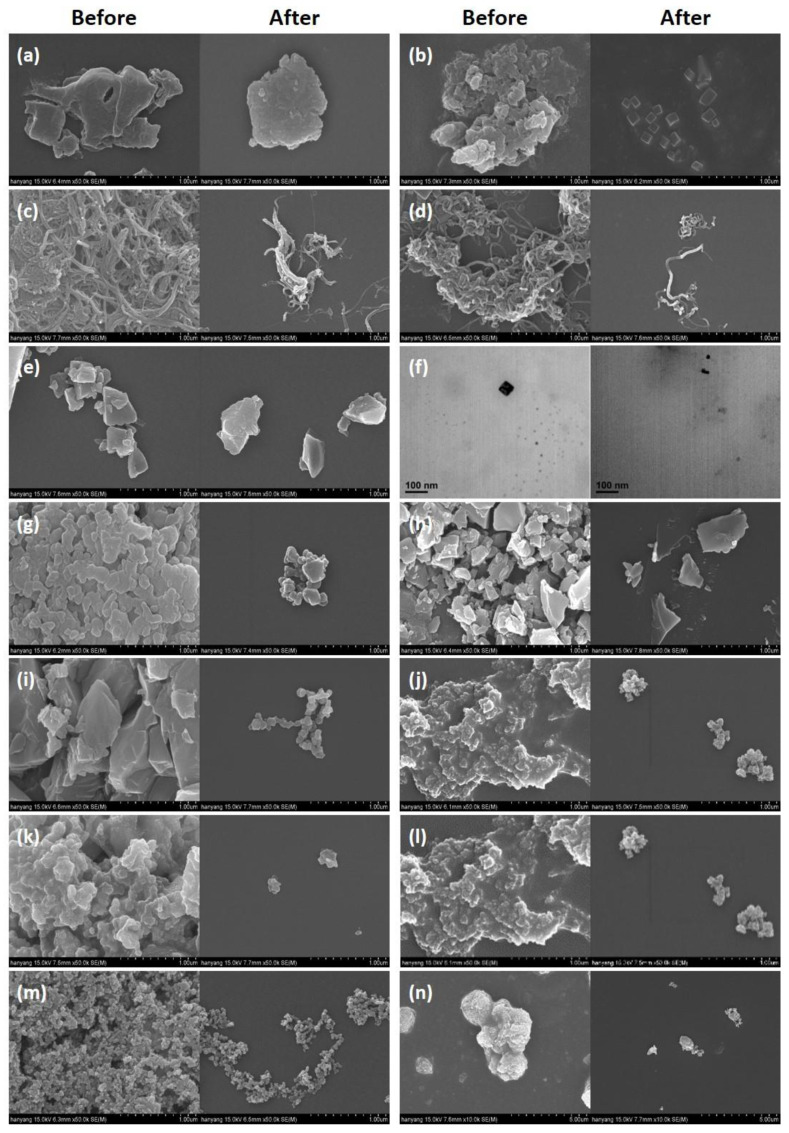
Microscopic images of manufactured nanomaterials before and after dispersion: (**a**) Al_2_O_3_, (**b**) carbon black, (**c**) SWCNTs, (**d**) MWCNTs, (**e**) CeO_2_, (**f**) dendrimers, (**g**) fullerene, (**h**) Au, (**i**) Fe, (**j**) nanoclays, (**k**) Ag, (**l**) SiO_2_, (**m**) TiO_2_, and (**n**) ZnO.

**Figure 5 ijerph-19-07140-f005:**
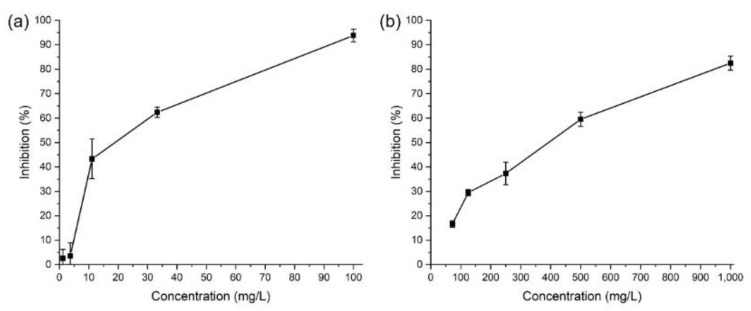
Growth inhibition of (**a**) SWCNTs and (**b**) TiO_2_ in 72 h.

**Table 1 ijerph-19-07140-t001:** Consumer uses and physical and chemical characteristics of manufactured nanomaterials [8,15,16,17,18,19,20,21,22,23].

Manufactured Nanomaterials	Use	Density (g/cm^3^)	Specific Surface Area (m^2^/g)	Solubility
Al_2_O_3_	Ceramic coating agent, ink additive, paint, catalyst	3.987	138	Insoluble
Carbon black	Tire, rubber reinforcing agent, paint, ink additive, filter	1.7	90–120	Insoluble
SWCNT	Coating agent, electrical material, photocatalyst base	1.3–1.4	700–900	Insoluble
MWCNT	Conductive filler, coating agent, solar cell, fuel cell	2.1	150–200	Insoluble
CeO_2_	Ceramic coating agent, abrasive agent	7.22	28	Insoluble
Dendrimers	Drug delivery, coating agent, catalyst carrier	0.813	N.A.	Soluble
Fullerene	Cosmetics	1.7–1.9	0.87	Insoluble
Au	Cosmetic additive, antibacterial agent, fuel cell, solar cell	19.3	48–59	Insoluble
Fe	Colorant, fuel cell catalyst, cell imaging, magnetic material	7.874	40–60	Insoluble
Nanoclays	Adsorbents, catalysts, coatings, filters	2.4	52	Insoluble
Ag	Antibacterial coating, water repellent coating, electrodes, conductive filler	10.49	18–22	Insoluble
SiO_2_	Paints, coatings, filters, insulation materials, LCD manufacturing, abrasives	2.1	189	Insoluble
TiO_2_	Cosmetics, paints, coatings, photocatalysts, solar cells	4.23	35–65	Insoluble
ZnO	Cosmetics, biosensors, coatings, transistors, solar cells	5.61	20–60	Insoluble

**Table 2 ijerph-19-07140-t002:** Toxicity test conditions using algae (OECD TG 201).

Test Parameter	Condition
Test species	*R. subcapitata*
Exposure method	Static
Experiment time (h)	72 (Measure after 24, 48, 72 h)
Temperature (°C)	23 ± 2
Intensity of light (lux)	6000 ± 1000
Photoperiod	Continuous lighting for 24 h
Size of chamber	250 mL Erlenmeyer flask
Volume of solution (mL)	100
Growth stage of test species	Exponentially growing stages
Initial inoculation conc. (cells/mL)	1 × 10^4^
Observation item (end point)	Cell density

**Table 3 ijerph-19-07140-t003:** Turbidity decrease rate of manufactured nanomaterials in 72 h.

Manufactured Nanomaterials	Turbidity Decrease Rate (%)		Manufactured Nanomaterials	Turbidity Decrease Rate (%)	
	100 rpm	200 rpm		100 rpm	200 rpm
Al_2_O_3_	27.7 ± 0.8	7.6 ± 0.6	Au	65.2 ± 5.1	10.4 ± 0.4
Carbon black	1.0 ± 0.01	1.1 ± 0.05	Fe	13.4 ± 0.6	4.7 ± 0.4
SWCNTs	11.0 ± 0.1	4.0 ± 0.05	Nanoclays	8.7 ± 0.3	−0.3 ± 0.01
MWCNTs	9.6 ± 0.8	8.8 ± 0.6	Ag	4.1 ± 0.2	3.6 ± 0.1
CeO_2_	−1.7 ± 0.1	−1.6 ± 0.04	SiO_2_	−0.7 ± 0.01	−0.2 ± 0.03
Dendrimers	0 ± 0.01	0 ± 0.03	TiO_2_	0.8 ± 0.02	−0.5 ± 0.01
Fullerene	11.3 ± 0.3	−2.8 ± 0.03	ZnO	8.3 ± 0.3	8.0 ± 0.2

**Table 4 ijerph-19-07140-t004:** Zeta potential of manufactured nanomaterials.

Manufactured Nanomaterials	Zeta Potential (mV)		Manufactured Nanomaterials	Zeta Potential (mV)	
	Cell Culture	With Gum Arabic		Cell Culture	With Gum Arabic
Al_2_O_3_	−13.7 ± 0.4	−20.7 ± 0.2	Au	−1.2 ± 0.03	−13.4 ± 0.3
Carbon black	−27.6 ± 2.8	−38.7 ± 0.5	Fe	−77.8 ± 8.9	−80.1 ± 7.8
SWCNT	−17.8 ± 0.9	−31.5 ± 0.8	Nanoclays	−30.6 ± 0.5	−35.8 ± 2.7
MWCNT	−16.1 ± 0.3	−32.9 ± 2.2	Ag	−37.1 ± 0.8	−40.8 ± 2.6
CeO_2_	−23.4 ± 0.7	−30.2 ± 2.1	SiO_2_	−32.0 ± 1.2	−39.5 ± 1.1
Dendrimers	NA	NA	TiO_2_	−15.8 ± 0.9	−29.7 ± 0.5
Fullerene	−25.9 ± 1.8	−36.5 ± 3.3	ZnO	28.6 ± 2.4	38.4 ± 0.2

## Data Availability

Not applicable.

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
