# Peer review of "Dispersion Stability of 14 Manufactured Nanomaterials for Ecotoxicity Tests Using Raphidocelis subcapitata"

_ijerph, 2022, doi:10.3390/ijerph19127140_

Round 1

Reviewer 1 Report

Attached please find the comments.

Author Response

Reviewer #1:

MINOR COMMENTS

  1. L 16. Delete “a” from the “a the results…”

We deleted “a” in the sentence:

“However, when manufactured nanomaterials are mixed with algae in a culture medium for ecotoxicity tests, the results are vulnerable to distortion by a coagulation phenomenon.”

  1. Abstract. Add all the chemicals abbreviations to harmonize the list of NPs used. Example; silver nanoparticles = AgNPs, multiwalled carbon nanotubes = MWCNTs, etc.

We added abbreviations of chemicals in the abstract and applied these throughout the text:

“Here, we describe a dispersion method commonly applicable to ecotoxicity tests for the 14 species of manufactured nanomaterials specified by the Organisation of Economic Co-operation and Development’s Sponsorship Programme, including aluminum oxide (Al2O3), carbon black, single-walled carbon nanotube (SWCNTs), multi-walled carbon nanotubes (MWCNTs), cerium oxide (CeO2), dendrimers, fullerene, gold (Au), iron (Fe), nanoclays, silver (Ag), silicon dioxide (SiO2), titanium dioxide (TiO2), and zinc oxide (ZnO).”

  1. L 90 and 91. The phrase is difficult to understand.

We revised the mentioned sentence to avoid the ambiguity:

“Manufactured nanomaterials to be dealt with in this study were selected by investigating the size and type of manufactured nanomaterials mainly used in industrial sites and research fields.”

  1. IN GENERAL, THE MANUSCRIPT’ GRAMMAR NEEDS TO BE REVIEWED AS ITS POOLY WRITTEN.

As commented, we corrected those awkward grammars and improper wordings with the help of native speakers. In addition, we can provide the certification (from Eworld editing)

  1. L 115. Delete “the” from “was the stopped for 40 s”

We deleted “the” and revised “was stopped for 40 s”:

“The shaking was stopped for 40 s before checking the culture medium’s dispersion stability.”

  1. L. 121. Authors mean preceding selection or preceding section?

Accordingly, we changed “preceding selection” by “preceding section”:

“The experiment described in the preceding section allowed us to select a dispersant that works effectively with the manufactured nanomaterials used in this study.”

  1. L. 149. Second “we” repeated.

We changed “We then” by “After then”:

“After then we transferred each of the culture media to a 250 mL Erlenmeyer flask, which was shaken for 72 h at 100–200 rpm in a shaking incubator (NB-205VQ, N-Biotek, Bucheon, Korea).”

  1. L. 171. Delete second been.

We deleted this:

“To verify the possibility of maintaining the dispersion stability, we conducted one such ecotoxicity test under experimental conditions that had already been obtained and performed an ecotoxicity test for SWCNTs and TiO2, for which ecotoxicity studies using R. subcapitata had been previously conducted.”

  1. L. 177. A common ratio below 3.2 … of what?

We revised the sentence:

“For the five treatment groups, excluding the control group, a common ratio below 3.2 at each nanomaterial concentration was maintained according to the OECD test method.”

  1. L. 269. Rephrase as “at such speed”

We changed “at such high speed” by “at such speed”:

“The result showed that, at such speed, algal growth after 72 h was 58 times the initial algae inoculation concentration.”

  1. L. 303. Zeta Arabic potential???

We changed “zeta arabic potential” by “zeta potential”:

“The result showed that the surface zeta potential of every material but ZnO was negative, and before adding gum arabic, all materials except Fe, nanoclays, and Ag showed a zeta potential below ±30 mV, maintaining an unstable or non-dispersion state.”

  1. Table 2 doesn’t exist in the manuscript.

We found the typo error in Table 2 and table number was changed by Table 4:

Table 4. Zeta potential of manufactured nanomaterials.”

  1. L. 325. Italicize R. subcapita

We changed to italicization:

“We are therefore confident in the validity of the ecotoxicity tests performed for SWCNTs and TiO2, for which ecotoxicity studies using R. subcapitata had been previously conducted, and we have found that our EC50 measurements were comparable with those reported in the existing literature.”

MAJOR COMMENTS

  1. It is recommended that authors deeply specify the election of each dispersant (consumer uses, where are found in nature, density differences, etc).

As referee’s comment, uses and properties of nanomaterials are added to Table 1.

  1. Clarify why the authors chose different concentrations for each ENMs.

The reason for the selection of different concentrations of manufactured nanomaterials was added:

“The concentration of manufactured nanomaterials was selected a concentration that could reach EC100 by referring to previous studies [15-22].”

“We selected and set the following concentrations of the manufactured nanomaterials at which we expected their respective culture media to exhibit a EC100: 500 mg/L for Al2O3, 1,000 mg/L for carbon black, 100 mg/L for SWCNTs, 500 mg/L for MWCNTs, 500 mg/L for CeO2, 100 mg/L for fullerene, 500 mg/L for Au, 500 mg/L for Fe, 1,000 mg/L for nanoclays, 20 mg/L for Ag, 500 mg/L for SiO2, 2,000 mg/L for TiO2, and 100 mg/L for ZnO.”

  1. Briefly summarize the OECD Test No. 201 to give a easy and general view of the ENMs preparation for readers.

Accordingly, A summary of OECD TG 201 is shown in Table 2, and related information has been added to the text:

“The conditions of all experiments was followed as OECD TG 201 (see Table 2)[27]. Also, the concentration of gum arabic was 750-12,000 mg/L and the common ratio of gum arabic concentration was 2.”

  1. Mat&Methods. Briefly indicate and describe the ecotoxicity test performed in R. subcapitata as section 2.7. Add statistics methodology as well if necessary.

We wrote the detail method of ecotoxicity test in Section 2.6 and added the description of statistics methodology:

“The half-maximal effective concentration (EC50) with a 95% confidence interval was calculated using the log-probit function in MedCalc (a toxicity-calculation software package) based on the growth inhibition rate, which was obtained through our own experimental efforts.”

  1. In section 2.3. Selection of sonication time, the authors do not specify the dispersion energy applied to the medium, which is a very important factor to take into account. Did the authors record the temperature along the time? Is well known that a big issue in NPs dispersion is the sample heating while sonicating, for that reason most of the protocols recommend placing the samples into an ice bath while sonicating. The heating can excite the NPs surface and create higher and irreversible NPs aggregates. Authors should deeply work in this considerations.

As referee’s comments, we added the detail information of experiment conditions such as temperature:

“The temperature of the culture medium was maintained at 20–25 ℃ using ice bath and the change in temperature was monitored throughout the whole experiment period.”

  1. Line 145. What the authors mean by “maximal effectiveness concentration”.

In the text, we changed “maximal effectiveness concentration” by “EC100” to avoid ambiguity.

  1. Line 211. Authors should not state that gum Arabic actually wraps around algae since they did not proof that action. There is no experiment where the authors show the gum doing so. The toxicity could come from other ways of action. A more careful sentence should be stated such as “Gum Arabic could potentially wrap around …. Since algal cells are… or summarize the finding of others in the text instead of just citing random literature.

Accordingly, we deleted to avoid the ambiguity.

  1. L. 215. Which phenomenon? Wrapping or being a nutrient source?

We deleted to avoid the ambiguity.

  1. Did the authors checked the oxygen consumption of the algae? Or the oxygen content of the medium. Once again, the authors are making statements that they have not proved. Be careful with how you deliver “hypotheses” vs “real findings”.

Accordingly, we deleted to avoid the ambiguity.

  1. Figure 1. How many experiments where performed? Is there any error bar plotted in those graphs. There was any statistical analysis performed along the manuscript? For a better visualization I would recommend that authors put the name of the NPs on top of each graph as a title. Moreover, graphical letters should be outside the graph on the top left.

We added the standard deviation in all Figures and Tables accordingly and graphical letters was shown on top left in figure.

  1. Figure 2. Again, I am lacking statistical analysis.

As referee’s comment, we added the standard deviation and could not conduct the statistical analysis due to low parameter.

  1. Lines 284-293. Why the authors call it coagulation and not agglomeration? I would suggest that this phenomenon occurs because of the long sonication times and as a consequence the sample heating.

We changed “coagulation” by “agglomeration” throughout whole text and added the detail information in session of sonication time:

“The temperature of the culture medium was maintained at 20–25 ℃ using ice bath and the change in temperature was monitored throughout the whole experiment period.”

  1. Lines. 294-295. From where did the authors extract this information. This would need to be cited.

We revised the mentioned sentence and cited the reference:

“Bae et al. [30] reported unique properties appeared through the electric double layer of manufactured nanomaterials. If the electric double layer is thick, the dispersion of manufactured nanomaterials in a solution is stable; if it is thin, agglomeration of nanomaterials can occur. The thickness of the electric double layer can be inferred by measuring the zeta potential. The closer the magnitude of the zeta potential is to zero, the thinner the electric double layer. Here, the magnitude of the zeta potential is affected by the composition of the solution, pH, ionic strength, and surface charge of the material. In general, it is known that aggregation occurs within ±5 mV of the zeta potential, dispersion is unstable within ±10 - ±30 mV, and dispersion is stably maintained when it is over ±30 mV [30].”

  1. Lines. 296-297. Authors should check and report in the manuscript the pH and Ionic strength of each solvent used as well as when with the 14 different NPs. Only then we could know if affected or not.

We added pH in solvents but there was little information about ionic strength:

“The pH of the culture medium was 6.4, and it was confirmed that it was maintained between 6.2 and 7.1 even after dispersion of the manufactured nanomaterial.”

  1. L. 297. Coagulation/aggregation occurs independently of the zeta potential. Is the measurement of the ZP what will tell you if your sample is stable or not. Authors should rephrase this sentence.

We revised the zeta potentials related sentence:

“In general, it is known that aggregation occurs within ±5 mV of the zeta potential, dispersion is unstable within ±10 - ±30 mV, and dispersion is stably maintained when it is over ±30 mV [30].”

  1. L. 300. The ZP analysis is not listed nor explained in the materials and methods. Moreover, a table summarizing all ZP results should be plotted in the manuscript to facilitate the interpretation of the findings. At which concentration did the authors measured the ZP. Is worth to mention that Nano Zetasizer machines have a NPs size and concentration limit. Out of there, measurements are wrongly made, interpreted and useless.

We revised the mentioned sentences:

“The concentration at which the toxicity of the manufactured nanomaterial is expected to represent EC100 was prepared as follows: 500 mg/L for Al2O3, 1,000 mg/L for carbon black, 100 mg/L for SWCNTs, 500 mg/L for MWCNTs, 500 mg/L for CeO2, 100 mg/L for fullerene, 500 mg/L for Au, 500 mg/L for Fe, 1,000 mg/L for nanoclays, 20 mg/L for Ag, 500 mg/L for SiO2, 2,000 mg/L for TiO2, and 100 mg/L for ZnO. The pH of the culture medium was 6.4, and it was confirmed that it was maintained between 6.2 and 7.1 even after dispersion of the manufactured nanomaterial.”

In the case of Brookhaven's ZetaPALS used in this study, the measurable size of nano materials is 1 nm-100 μm. Thus, there is no problem in using it in the size range of the manufactured nanomaterials. Also, since the maximum concentration is 40%, there is little effect by concentration.

  1. Lines 313 to 315 should be deleted as increments or reductions of ZP are not proportional.

We modified the mentioned reference and described in Table 4:

“In fact, the absolute value of zeta potential increased by 118.1% on average compared with zeta potentials before the addition of gum arabic.”

  1. Table one is repeated in page 8 and 9.

We changed the second Table 1 by Table 2.

  1. Rephrase figure 4 and add scale bars to each image or mention in the figure captions.

Since the scale bar of the existing Figure 4 is too small to be seen, we resized the figure to make the scale bar more visible.

  1. Why the authors only checked the ecostoxicity of 2 NMs? Should be interesting to check all the rest as for example zinc and silver nanoparticles are very toxics in a wide range on living organisms.

The ecotoxicity test results for 12 types of manufactured nanomaterials except SWCNTs and TiO2 were added by authors’ another paper:

“Authors reported the EC50 measured by the same dispersion method [37]: Al2O3: 53.9 ± 10.2 mg/L, carbon black: 4.8 ± 2.2 mg/L, MWCNTs: 9.0 ± 1.5 mg/L, CeO2: 26.2 ± 4.6 mg/L, dendrimers: 24.6 ± 3.9 mg/L, fullerene: 395.3 ± 51.6 mg/L, Au: 352.6 ± 137.3 mg/L, Fe: 64.3 ± 10.6 mg/L, nanoclays: 11,245.6 ± 2,342.3 mg/L, Ag: 0.3 ± 0.04 mg/L, SiO2: 211.3 ± 22.7 mg/L, and ZnO: 0.06 ± 0.04 mg/L.

  1. Did the authors analyze the ionic content of those NMs known to have a high dissociation rate when in solution (zinc and silver).

Since the main purpose of this study is to investigate the stability of dispersion, we could not cover the mechanism of toxicity including ionic concentrations of nanomaterials such as Zn and Ag.

  1. CONCLUSIONS. FIRST SENTENCE. From what I have been reading along this manuscript, the aim of this manuscript is determining the suspension stability of 14 ENMs, and only two toxicity tests were performed … then this study DID NOT aimed to establish any toxicity test method.

As referee’s comment, we revised conclusions session:

“This study aimed to investigate dispersion stability of manufactured nanomaterials in culture for the toxicity test method using R. subcapitata as the OECD official test species.” 

Reviewer 2 Report

Dear Editor and Authors

The manuscript brings interesting results from a methodological point of view. Although the methodology is related to a protocol for carrying out ecotoxicological tests, there is a lack of greater emphasis on the results derived from tests with algae. In this sense, there is no specific topics to describe the methodologies and results and discussion of tests performed with algae.

Another point of concern is that the title suggests that ecotoxicological tests were carried out with 13 nanomaterials. However, tests were performed with 2 nanomaterials and the dispersion method, necessary for carrying out the OECD protocol, was carried out with all nanomaterials.

Therefore, I believe that the manuscript does not fully fit within the scope of IJERPH. I suggest that the manuscript be transferred to another MDPI journal.

Minor comments:

Line 16: Delete “a” before “the results”

There is no standard deviation in the results of Figs. 1 and 3?

Author Response

Reviewer #2:

The manuscript brings interesting results from a methodological point of view. Although the methodology is related to a protocol for carrying out ecotoxicological tests, there is a lack of greater emphasis on the results derived from tests with algae. In this sense, there is no specific topics to describe the methodologies and results and discussion of tests performed with algae.

Another point of concern is that the title suggests that ecotoxicological tests were carried out with 13 nanomaterials. However, tests were performed with 2 nanomaterials and the dispersion method, necessary for carrying out the OECD protocol, was carried out with all nanomaterials.

This study was conducted to investigate dispersion stability conditions for manufactured nanomaterials, which should be given the highest priority in order to conduct an ecotoxicity test of manufactured nanomaterials using algae. To this end, experiment conditions required for dispersing manufactured nanomaterials, including dispersants, have been considered. In addition, since we aimed to ensure dispersion stability for ecotoxicity experiments using algae, all of the effects on algae were tested for dispersant concentration and stirring RPM that could directly or indirectly affect algae in each dispersion condition.

This study was conducted to solve the problem of not being able to compare the toxicity of each material because the existing ecotoxicity studies on manufactured nanomaterials have different dispersion conditions and test organisms. This research result, which can be applied uniformly to the dispersion of manufactured nanomaterials, is considered to be a reference data for ecotoxicity tests on 14 types of manufactured nanomaterials, as well as other manufactured nanomaterials, which are the priority research targets of the OECD.

Therefore, I believe that the manuscript does not fully fit within the scope of IJERPH. I suggest that the manuscript be transferred to another MDPI journal.

Minor comments:

Line 16: Delete “a” before “the results”

We deleted “a” in the sentence:

“However, when manufactured nanomaterials are mixed with algae in a culture medium for ecotoxicity tests, the results are vulnerable to distortion by a coagulation phenomenon.”

There is no standard deviation in the results of Figs. 1 and 3?

We added the standard deviation in all Figures and Tables accordingly 

Reviewer 3 Report

Paper title: “Dispersion Stability in Ecotoxicity Tests of 13 Manufactured Nanomaterials Using Algae” by Lee et al. describes the ecotoxicity of 14 nanomaterials when mixed with algal culture. Authors investigate the efficacy of different parameters in the dispersion of nanomaterials such as selection of optimum dispersant, Selection of sonication time, Selection of stirring speed, Measurement of shape and size distribution, and Evaluation of dispersion stability. The obtained data are promising and can be considered for publication after major revision.  

  • The title should be rephrased, I understand from the current title that the 13 nanomaterials were synthesized by algae, but this does not correct.
  • In the title, the authors said “… of 13 Manufactured…”, while in the whole manuscript, the authors mentioned the toxicity was achieved for 14 manufactured nanomaterials, please check.
  • Line 43 and 44: Keller et al. (2010).., Sillanpää et al. (2011); should be written as follows: Keller et al…., Sillanpää et al….
  • Please check the reference style throughout the manuscript, such as lines 50, 59, 61, ….
  • Line 61: “sonification” or “sonication” please check.
  • Line 67: reference was written one as a number and one as author name, please check.
  • A clear hypothesis or aim of the work should be added at the end of the introduction section.
  • Line 102: “The three reagents we used as” should be “The three reagents were used as”
  • The term “We” has been repeated a lot in the manuscript, please avoid this repetition as possible.
  • What about the control in section 3.1. Selection of dispersants (effect of dispersant without nanomaterials)?
  • The obtained data should be statistically analyzed and adding error bars for each point in the figures.
  • Deep discussion is needed in the synthetic methods and validation techniques.
  • In Figure 2, error bars were added only at some point, why? Please check.
  • Data in the Tables should be statistically analyzed and adding ±SE or ±SD
  • Table 1 is repeated twice, on pages 8 and 9 with the same numbers, please check.
  • The conclusion should be rephrased to be concise.
  • Extensive editing of the English language and style is required

Author Response

Reviewer #3:

Paper title: “Dispersion Stability in Ecotoxicity Tests of 13 Manufactured Nanomaterials Using Algae” by Lee et al. describes the ecotoxicity of 14 nanomaterials when mixed with algal culture. Authors investigate the efficacy of different parameters in the dispersion of nanomaterials such as selection of optimum dispersant, Selection of sonication time, Selection of stirring speed, Measurement of shape and size distribution, and Evaluation of dispersion stability. The obtained data are promising and can be considered for publication after major revision. 

  1. The title should be rephrased, I understand from the current title that the 13 nanomaterials were synthesized by algae, but this does not correct.

As referee’s suggestion, we changed “13” by “14” in title.

  1. In the title, the authors said “… of 13 Manufactured…”, while in the whole manuscript, the authors mentioned the toxicity was achieved for 14 manufactured nanomaterials, please check.

As referee’s suggestion, we changed “13” by “14” in title.

  1. Line 43 and 44: Keller et al. (2010).., Sillanpää et al. (2011); should be written as follows: Keller et al…., Sillanpää et al….

We revised the reference format throughout the whole text.

  1. Please check the reference style throughout the manuscript, such as lines 50, 59, 61, ….

We revised the reference format throughout the whole text.

  1. Line 61: “sonification” or “sonication” please check.

We changed “sonification” by “sonication”:

“For example, Hartmann et al. [10] conducted sonication for 10 min immediately before dispersing a 250 ppm TiO2 suspension, kept it in a dark place at 5 ℃ until the test was finished, then conducted sonication again for 10 min.”

  1. Line 67: reference was written one as a number and one as author name, please check.

Accordingly, we deleted the mentioned reference format.

  1. A clear hypothesis or aim of the work should be added at the end of the introduction section.

We revised the purpose of research:

“In the toxicity test of other manufactured nanomaterials not covered in this study, it may be a reference material that can solve the problem of dispersion.”

  1. Line 102: “The three reagents we used as” should be “The three reagents were used as”

The term “We” has been repeated a lot in the manuscript, please avoid this repetition as possible.

Accordingly, we changed “The three reagents we used as” by “The three reagents were used a”.

  1. What about the control in section 3.1. Selection of dispersants (effect of dispersant without nanomaterials)?

For the control group, the difference in turbidity for each manufactured nanomaterial were monitored under the condition that no dispersant was added was used. The test results for the control group are shown at the 0 mg/L point in Figure 1.

  1. The obtained data should be statistically analyzed and adding error bars for each point in the figures.

As referee’s comment, we added the standard deviation in all figures and could not conduct the statistical analysis due to low parameter.

  1. Deep discussion is needed in the synthetic methods and validation techniques.

We did not add the synthetic method of manufactured nanomaterials because they were purchased and used without directly synthesizing them.

  1. In Figure 2, error bars were added only at some point, why? Please check.

In the case of two points where the error bar of Figure 2 is not visible, the error is too small to be displayed well on the graph.

  1. Data in the Tables should be statistically analyzed and adding ±SE or ±SD

We added the standard deviation in all Tables.

  1. Table 1 is repeated twice, on pages 8 and 9 with the same numbers, please check.

We changed the second Table 1 by Table 2.

  1. The conclusion should be rephrased to be concise.

We revised conclusions:

“This study aimed to investigate dispersion stability of manufactured nanomaterials in culture medium for the toxicity test method using R. subcapitata as the OECD official test species. However, for toxicity tests using R. subcapitata as specified in OECD Test No. 201, a uniform dispersion method for manufactured nanomaterials is required because agglomeration and precipitation of the manufactured nanomaterials can occur due to the effect of cations present in the culture medium. To this end, the optimal dispersant and its injection concentration, the optimal sonication time, and the optimal stirring speed must be carefully chosen and calibrated. First, as for the optimal dispersant and its injection concentration, 1,500 mg/L of gum arabic was selected in consideration of the dispersing performance and the effect of the dispersant on algae. For the optimal sonication time, our test results indicated that a different sonication time should be used for each material. We found that the optimal dispersion efficiency was obtained at 1 h for dendrimers, 2 h for SiO2, 24 h for SWCNT and Au, and 4 h for the rest of the OECD-listed manufactured nanomaterials, which suggests that these values are widely applicable. Regarding the optimal stirring speed, we found that dispersion stability can be maintained for 72 h when the material is stirred at 200 rpm. We also verified that dispersion stability can be achieved by changing the zeta potential of the manufactured nanomaterials. The zeta potential increased by 118.1% on average compared to before dispersion treatment, confirming the stability of dispersion. The optimal dispersion conditions for ecotoxicity tests of manufactured nanomaterials, and the proposed dispersion method for them, should be applicable to ecotoxicity tests for manufactured nanomaterials other than the 14 species listed in the OECD’s Sponsorship Programme for the Testing of Manufactured Nanomaterials.”

  1. Extensive editing of the English language and style is required

As commented, we corrected those awkward grammars and improper wordings with the help of native speakers. In addition, we can provide the certification (from Eworld editing) 

Reviewer 4 Report

The manuscript as a whole is well written and contains interesting data.
However, a very important point which needs to be clarified by the authors
concerning the materials and methods part. Only in Figures 2 and 5 the results
have been presented in the form of mean with standard deviation.
On the other hand, in the materials and methods section, the authors did not
indicate how many times the manipulations were repeated? Why was the other data
not presented as the mean with standard deviation? A statistical study to compare
all these data is missing.
The conclusion needs to be shortened and focused only on the main results obtained.So no need to repeat data that has already been well described in the results and discussion section. Other remarks underlined in yellow in the attached file also need to be corrected before publication.

Author Response

Reviewer #4:

  1. The manuscript as a whole is well written and contains interesting data. However, a very important point which needs to be clarified by the authors concerning the materials and methods part.

We revised “materials and methods’ session to clarify ambiguity and added Table 2:

“The conditions of all experiments was followed as OECD TG 201 (see Table 2)[27]. the concentration of gum arabic was 750-12,000 mg/L and the common ratio of gum arabic concentration was 2.”

“Table 2. Toxicity test conditions using algae (OECD TG 201).”

“The concentration at which the toxicity of the manufactured nanomaterial is expected to represent EC100 was prepared as follows: 500 mg/L for Al2O3, 1,000 mg/L for carbon black, 100 mg/L for SWCNTs, 500 mg/L for MWCNTs, 500 mg/L for CeO2, 100 mg/L for fullerene, 500 mg/L for Au, 500 mg/L for Fe, 1,000 mg/L for nanoclays, 20 mg/L for Ag, 500 mg/L for SiO2, 2,000 mg/L for TiO2, and 100 mg/L for ZnO. The pH of the culture medium was 6.4, and it was confirmed that it was maintained between 6.2 and 7.1 even after dispersion of the manufactured nanomaterial.”

  1. Only in Figures 2 and 5 the results have been presented in the form of mean with standard deviation. On the other hand, in the materials and methods section, the authors did not indicate how many times the manipulations were repeated? Why was the other data not presented as the mean with standard deviation? A statistical study to compare all these data is missing.

We added the standard deviation to all tables and figures. Also, the repeat experiment contents were added to confirm reproducibility:

“The dispersion stability was determined as the average value of three repeated measurements of the difference in turbidity between the upper and lower layers (1/3 and 2/3) of the graduated cylinder using a turbidimeter (2100Q, HACH Co., USA).”

“10 mL was collected from the upper and lower layers for each sonication time, and the difference in turbidity was measured three times.”

“Because collecting samples from a 250 mL Erlenmeyer flask is difficult due to low height of the solution is low (leaving a small distance between the top and bottom layers of the solution), it is crucial to verify dispersion stability by measuring turbidity three times hourly at a half point of the solution and then comparing it with the initial turbidity.”

  1. The conclusion needs to be shortened and focused only on the main results obtained. So no need to repeat data that has already been well described in the results and discussion section.

As referee’s comment, we revised the conclusion session:

“This study aimed to investigate dispersion stability of manufactured nanomaterials in culture medium for the toxicity test method using R. subcapitata as the OECD official test species. However, for toxicity tests using R. subcapitata as specified in OECD Test No. 201, a uniform dispersion method for manufactured nanomaterials is required because agglomeration and precipitation of the manufactured nanomaterials can occur due to the effect of cations present in the culture medium. To this end, the optimal dispersant and its injection concentration, the optimal sonication time, and the optimal stirring speed must be carefully chosen and calibrated. First, as for the optimal dispersant and its injection concentration, 1,500 mg/L of gum arabic was selected in consideration of the dispersing performance and the effect of the dispersant on algae. For the optimal sonication time, our test results indicated that a different sonication time should be used for each material. We found that the optimal dispersion efficiency was obtained at 1 h for dendrimers, 2 h for SiO2, 24 h for SWCNT and Au, and 4 h for the rest of the OECD-listed manufactured nanomaterials, which suggests that these values are widely applicable. Regarding the optimal stirring speed, we found that dispersion stability can be maintained for 72 h when the material is stirred at 200 rpm. We also verified that dispersion stability can be achieved by changing the zeta potential of the manufactured nanomaterials. The zeta potential increased by 118.1% on average compared to before dispersion treatment, confirming the stability of dispersion. The optimal dispersion conditions for ecotoxicity tests of manufactured nanomaterials, and the proposed dispersion method for them, should be applicable to ecotoxicity tests for manufactured nanomaterials other than the 14 species listed in the OECD’s Sponsorship Programme for the Testing of Manufactured Nanomaterials.”

  1. Other remarks underlined in yellow in the attached file also need to be corrected before publication.

We corrected the underlined part:

1) 13 → 14

2) Algae → Raphidocelis subcapitata

3) We deleted “a” in the sentence:

“However, when manufactured nanomaterials are mixed with algae in a culture medium for ecotoxicity tests, the results are vulnerable to distortion by a coagulation phenomenon.”

4) We double-checked figure:

“Here, we describe a dispersion method commonly applicable to ecotoxicity tests for the 14 species of manufactured nanomaterials specified by the Organisation of Economic Co-operation and Development’s Sponsorship Programme”

5) targeting → using

6) “algae”  deleted

7) species → types

8) Keller et al. (2010) → Keller et al. [4]

9) Sillanpää et al. (2011) → Sillanpää et al. [5]

10) We deleted the reference number.

11) Pan et al. (2012) → Pan et al. [7]

12) We deleted the reference number.

13) to Scrobicularia plana bivalves → to Scrobicularia plana

14) Hartmann et al. (2010) → Hartmann et al. [10]

15) We deleted the reference number.

16) Hartmann et al. (2010) → Hartmann et al. [11]

17) We deleted the reference number.

18) (Li et al., 2010) → deleted

19) Knauer et al. (2007) → Knauer et al. [13]

20) on Raphidocelis subcapitata algae → on Raphidocelis subcapitata

21) We deleted the reference number.

22) Canesi et al. (2010) → Canesi et al. [14]

23) used Mytilus galloprovincialis mussels → used Mytilus galloprovincialis

24) species → model

25) We changed “We then” by “After then”:

“After then we transferred each of the culture media to a 250 mL Erlenmeyer flask, which was shaken for 72 h at 100–200 rpm in a shaking incubator (NB-205VQ, N-Biotek, Bucheon, Korea).”

26) We changed “in the algae culture medium and the algae culture medium with gum arabic added” by “in the algal culture medium with and without the gum arabic”

27) algae → R. subcapitata

28) algae → alga

29) van der → Van Der

31) “R. subcapitata” was changed to italicize “R. subcapitata

Reviewer 5 Report

Dear author,

Thanks for the effort put into this manuscript, clearly appreciated from all perspectives: the careful writing, the clarity of the ideas and explanations, the logic research design, and the robust conclusions.

Anyway, please consider a few minor recommendations in order to improve this already great manuscript.

First, two issues concerning the SEM images in the Figure 4. On one side, the image size is so small that it is quite difficult to appreciate whether they are comparable or not. Besides, there is no reference to this. Anyhow, rising the images size would clearly improve this figure, even if it requires a single page. Then, please consider to include a comparable size bar, even if it might only be interesting for the series of 3 images.

On the other side, a question arising from my lack of understanding this: why 3 images per series? I understand before and after the treatment. But the other one? In the text, you said “The shapes of each material before and after treatment are summarized in Fig.4”. And the caption says “Microscopic image of manufactured nanomaterials ABOUT BEFORE and after dispersion: …”. I do not understand very much what you meant with this caption, but anyway I expected two images per series, and do not know why you included 3. Maybe this is clear and I do not see at this moment, but maybe a better explanation makes it easier to understand for those readers far from your field.

Within the “evaluation of dispersion stability” part of the manuscript, I would improve the explanations why counting on a greater EC50 that those previously obtained might be of interest to defend your method to disperse and preserve the stability of that dispersion, at least through the ecotoxicity test (if the nanomaterial is toxic and the dispersion is stable, it should promote a higher EC50 that those tests where the nanomaterial suffers from coagulation, and the nanomaterial is not available to all bioreagents within the ecotoxicity test). This would strengthen the conclusions, which indeed are aligned with this idea.

Indeed, my final recommendation would be strengthening the conclusions, by including a sentence making reference to the obtention of a reproducible and versatile method to get dispersion stability for the 14 OECD-mentioned species applicable to ecotoxicity tests, between the two final ideas:  the BSA explanation, and the applicability to other species… This way you will make the reader know you made it, both for the OECD-mentioned species, and it might also work for the rest of them.

Finally, minor language corrections:

  1. Abstract, line 16. “a the results”
  2. Section 2.2, line 115. “The shaking was the stopped for 40s…”
  3. Section 3.2, line 236. “Sonification”

Best luck,

Author Response

Reviewer #5:

Thanks for the effort put into this manuscript, clearly appreciated from all perspectives: the careful writing, the clarity of the ideas and explanations, the logic research design, and the robust conclusions. Anyway, please consider a few minor recommendations in order to improve this already great manuscript.

  1. First, two issues concerning the SEM images in the Figure 4. On one side, the image size is so small that it is quite difficult to appreciate whether they are comparable or not. Besides, there is no reference to this. Anyhow, rising the images size would clearly improve this figure, even if it requires a single page. Then, please consider to include a comparable size bar, even if it might only be interesting for the series of 3 images.

Since the scale bar of the existing Figure 4 is too small to be seen, we resized the figure to make the scale bar more visible.

We added the detail description:

“The shapes of each material before and after treatment are summarized in Fig. 4. The sizes of manufactured nanomaterials increased by an average factor of 21 compared with their sizes before treatment.”

  1. On the other side, a question arising from my lack of understanding this: why 3 images per series? I understand before and after the treatment. But the other one? In the text, you said “The shapes of each material before and after treatment are summarized in Fig.4”. And the caption says “Microscopic image of manufactured nanomaterials ABOUT BEFORE and after dispersion: …”. I do not understand very much what you meant with this caption, but anyway I expected two images per series, and do not know why you included 3. Maybe this is clear and I do not see at this moment, but maybe a better explanation makes it easier to understand for those readers far from your field.

According to referee's opinion, the microscopic image before the addition of dispersant into the culture medium was judged to be unnecessary, so three related images was deleted and the size of Fig. 4 increased.

  1. Within the “evaluation of dispersion stability” part of the manuscript, I would improve the explanations why counting on a greater EC50 that those previously obtained might be of interest to defend your method to disperse and preserve the stability of that dispersion, at least through the ecotoxicity test (if the nanomaterial is toxic and the dispersion is stable, it should promote a higher EC50 that those tests where the nanomaterial suffers from coagulation, and the nanomaterial is not available to all bioreagents within the ecotoxicity test). This would strengthen the conclusions, which indeed are aligned with this idea.

EC50 is a half maximal effective concentration, and the lower this concentration, the stronger the toxicity. In the case of BSA, which is mainly used as a dispersant in the ecotoxicity study of existing manufactured nanomaterials, it formed BSA-nanomaterial and showed a higher EC50 measurement value than the dispersion method used in this study. In fact, in this study, it was confirmed that the toxicity was about 20-40% higher than the measurement result using the conventional BSA. Through this, it was determined that the existing dispersant, BSA, could be substituted.

  1. Indeed, my final recommendation would be strengthening the conclusions, by including a sentence making reference to the obtention of a reproducible and versatile method to get dispersion stability for the 14 OECD-mentioned species applicable to ecotoxicity tests, between the two final ideas: the BSA explanation, and the applicability to other species… This way you will make the reader know you made it, both for the OECD-mentioned species, and it might also work for the rest of them.

The ecotoxicity test results for 12 types of manufactured nanomaterials except SWCNTs and TiO2 were added by authors’ another paper:

We added the added the related sentence and cited authors’ another paper:

“Authors reported the EC50 measured by the same dispersion method [37]: Al2O3: 53.9 ± 10.2 mg/L, carbon black: 4.8 ± 2.2 mg/L, MWCNTs: 9.0 ± 1.5 mg/L, CeO2: 26.2 ± 4.6 mg/L, dendrimers: 24.6 ± 3.9 mg/L, fullerene: 395.3 ± 51.6 mg/L, Au: 352.6 ± 137.3 mg/L, Fe: 64.3 ± 10.6 mg/L, nanoclays: 11,245.6 ± 2,342.3 mg/L, Ag: 0.3 ± 0.04 mg/L, SiO2: 211.3 ± 22.7 mg/L, and ZnO: 0.06 ± 0.04 mg/L.

  1. Finally, minor language corrections:

We corrected the typo error according to referee’s comment:

1) Abstract, line 16. “a the results”

We deleted “a” in the sentence:

“However, when manufactured nanomaterials are mixed with algae in a culture medium for ecotoxicity tests, the results are vulnerable to distortion by a coagulation phenomenon.”

2) Section 2.2, line 115. “The shaking was the stopped for 40s…”

We revised the sentence:

“The shaking was stopped for 40 s before checking the culture medium’s dispersion stability.”

  1. Section 3.2, line 236. “Sonification”

We changed “sonification” by “sonication” throughout whole text.

Round 2

Reviewer 1 Report

After revising the author's responses I am still not sure that this manuscripts fulfill the scope of IJERPH. The reasons are the following: (1) There are no statistical analysis along the manuscript; (2) I am not sure that MNPs characterization was properly done (more concrete when analyzing ZP); and last (3) authors have not responded nor discussed any of the doubts, concerns or questions I had related to the experimental design, results or discussion. Authors just eliminated the more controversial or ambiguous comments in the manuscript instead of trying to clarify or improve it.

Author Response

  1. There are no statistical analysis along the manuscript.

In this experiment, the amount of test solution is 100 mL, and if it is used for analysis, the solution will be consumed to the extent that the analysis is impossible in the last 72 hours. That is because this study was conducted to immediately apply the dispersion method to the ecotoxicity test. Accordingly, 3 sets of samples were prepared and the experiment was conducted, and the average and standard deviation were calculated. As a result, additional statistical analysis is meaningless due to the small number of populations, but it is a very common situation in probability because the experiment was conducted with enough samples stirred. If there is any additional statistical analysis method you would like in this study, please let us know. We'll apply it right away and conduct a statistical analysis.

  1. I am not sure that MNPs characterization was properly done (more concrete when analyzing ZP).

According to referee’s comment, we added characteristics of manufactured nanomaterials such as specific surface to Table 1 and demonstrated references.

  1. Authors have not responded nor discussed any of the doubts, concerns or questions I had related to the experimental design, results or discussion. Authors just eliminated the more controversial or ambiguous comments in the manuscript instead of trying to clarify or improve it.

We tried to clarify the statement about the cause of the effect of gum arabic on algae. However, as reviewer’s opinion, there was no experimental result for toxicity of gum arabic after a peer literature review. Therefore, we deleted the mentioned paragraph based on our improper hypothesis.

Reviewer 2 Report

No comments.

Author Response

Since there is a comment of referee, we don't have any response.

Reviewer 3 Report

-       Still the title needs modification. The current title means the 14 nanomaterials were synthesized by Raphidocelis subcapitata and this is not correct. The Raphidocelis subcapitata was used as a model to investigate the toxicity of 14 manufactured nanomaterials.

-       Line 84 and 85: the statement should be rephrased to be clearer.

-       Please check the error bars in figure 3.

-       Extensive editing of English language and style required

Author Response

  1. Still the title needs modification. The current title means the 14 nanomaterials were synthesized by Raphidocelis subcapitata and this is not correct. The Raphidocelis subcapitata was used as a model to investigate the toxicity of 14 manufactured nanomaterials.

Accordingly, we revised tile:

“ Dispersion Stability of 14 Manufactured Nanomaterials for Ecotoxicity Test Using Raphidocelis subcapitata

  1. Line 84 and 85: the statement should be rephrased to be clearer.

According to referee’s comment, we changed “ “In the toxicity test of other manufactured nanomaterials not covered in this study, it may be a reference material that can solve the problem of dispersion” by “These results can be helpful in securing dispersion conditions for the toxicity test of various manufactured nanomaterials”.

  1. Please check the error bars in figure 3.

We revised invisible error bar was enlarged and inserted into the original figure.

  1. Extensive editing of English language and style required

According to referee’s comment, we we corrected those awkward grammars and improper wordings with the help of native speakers. In addition, we can provide the certification (from Eworld editing)

Reviewer 4 Report

Thanks to the authors for responding clearly to my comments.
I consider that the manuscript is now publishable in this form

Author Response

(The authors gave the same response as above.)
